# The Impact of Mitigating Circumstances Procedures: Student Satisfaction, Wellbeing and Structural Compassion on the Campus

**Neil Armstrong \* and Nicola C. Byrom** 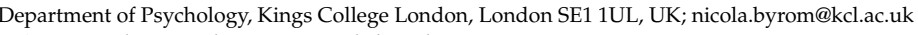

Department of Psychology, Kings College London, London SE1 1UL, UK; nicola.byrom@kcl.ac.uk
\* Correspondence: neil.1.armstrong@kcl.ac.uk

**Abstract:** For several decades, universities have sought to promote better mental health amongst students. For example, universities now have formal institutional arrangements to postpone deadlines for students where there are mitigating circumstances, such as ill health. Such provisions might be understood as praiseworthy cases of institutional compassion. But, empirical research is needed to investigate how these measures play out in practice. This paper draws on ethnographic research in several UK universities to explore the experiences of staff responsible for the enactment of mitigating circumstance provisions. We find staff members sympathetic to the aims of the measures but also sceptical, and in some cases angry, because they find that the provisions have unwanted and undesirable effects. This paper uses the wider social science literature on bureaucracy to consider why this might be the case and raises questions about the capacity of institutions to enact ethical ideals.

**Keywords:** student mental health; mitigating circumstances; disability; whole university approach; care; bureaucracy; help-seeking; post-COVID-19; community

## 1. Introduction

There is a long history of UK universities taking an interest in the mental health of their students [1]. Mental health services were developed by universities for students in the decades after the second world war, with University College Leicester offering services in 1948 and the London School of Economics offering services in 1952 (pp. 206–207). Over the past decade, the sector has seen a transition from focusing on reactive and individualistic measures towards an institution-wide or whole university approach. This shift can be seen in the Step Change framework introduced by Universities UK and the University Mental Health Charter published by the student mental health charity Student Minds [2,3]. This approach has been motivated by an increasing understanding of the importance of context and a recognition that an individual's mental health is influenced by a wide range of societal and environmental factors [4]. The University Mental Health Charter asks universities to take a multi-stranded approach, considering how all aspects of university life, including studying and social life, influence mental health [3]. The whole university approach shifts the focus from the individual to the institution, from treatment to prevention and from mental disorder to wellbeing. It promotes change at a cultural level, meaning that values such as 'care' and 'compassion' are sedimented into the university, not incidental or added on, but close to the heart of how the university functions. In this paper, we call this goal 'structural compassion' or 'structural care'.

Terms like 'compassion' and 'care' have increasing currency in discussions of civic life, but, as Joshua Hordern points out, they can be difficult to pin down [5]. Lauren Berlant understands compassion in relational terms, as 'a technology of belonging' [6]. She suggests that compassion leads to 'a more nuanced and capacious engagement with the scenes of human activity' (p. 9). This suggests that a structurally compassionate university might feature better relationships between its members. Mitigating circumstance

arrangements might be seen as instances of structural compassion in that they refer to engagement between staff and students. If the arrangements are working well, they change the relationship between student and institution, such that, in Berlant's terms, a more nuanced and capacious engagement is enabled.

Improved relationships between individual students and individual members of staff are likely to have wider, systemic effects. Recent work on the ethics of care by philosophers such as Carol Gilligan helps provide an account of these processes [7]. Gilligan argues that caring relationships promote empathy, sensitivity, trust and responsiveness, and these effects radiate out from particular instances of care. Compassion emerges as humane and life-enhancing, promoting desirable qualities in the individual and connecting communities by means of their common humanity. Jonathan Haidt suggests that communities that value compassion and empathy promote individual and social flourishing [8]. A structurally caring campus might be expected to be healthier and happier, a community that helps students to thrive.

Structural compassion on the campus is a holistic enterprise, a cultural goal that is complex and multifaceted. It refers not just to systems and processes but also to the vibrancy (or otherwise) of communities amongst university populations and to personal behaviours, impressions, values and orientations. It is an open question as to how these goals might be achieved. We might be sceptical of attempts to remove systemic biases in organisations through rules and regulations if staff members themselves remain biased. The Francis inquiry into the catastrophic failings of the mid-Staffordshire NHS found that staff treated patients 'with what appeared to be callous indifference' for a number of reasons, including a culture that 'was not conducive to providing good care for patients or providing a supportive working environment for staff' but also that 'there was an atmosphere of fear of adverse repercussions; a high priority was placed on the achievement of targets'. In other words, alongside cultural failings, accountability practices that might be expected to promote care quality became a barrier to care quality [9]. Thinking about the university context, we might expect that a transition to a compassionate campus will require a range of measures, encompassing policy, pedagogy, training and bureaucratic instruments, and we might note that these measures can be both part of the solution and, if misjudged or dysfunctional, part of the problem.

One of the pillars of the University Mental Health Charter focuses on learning. While students have variable levels of contact with student services, all students connect with their teachers, their pedagogy, and the course curriculum and assessment processes. The Charter urges universities to ensure that curriculum takes a holistic and inclusive view of learners, using evidence informed practice and secure pedagogic scaffolding to enable all students to develop skills, confidence and academic self-efficacy and to improve performance [10–13].

The inclusive education approach, advocated by the University Mental Health Charter, extends from the Equalities Act (2010). The Act prohibits direct or indirect discrimination on the basis of disability, including mental health, placing a duty on universities to ensure that their teaching and education practices do not discriminate against individuals based on mental health. To comply with this, universities are required to make reasonable adjustments to ensure that students with mental health problems have equal access to education. Initial responses to this act might have included providing alternative formats for course materials, adjusting assessment methods and providing specific allowances for the ways students attend classes. Universities have complied with the Equalities Act 2010 through the provision of specific disability services and, at times, additional support funded by the Disabled Students Allowance. Students are required to register with the disability service, formally disclosing their mental health problems, such that the service might develop a tailored education plan to be shared with their academic department.

Mitigating circumstance arrangements are an instance of universities attempting to make reasonable adjustments to disability. They are administrative provisions that make it possible for student work deadlines to be delayed because of pressing health needs. Mitigating circumstance arrangements are bureaucratic instruments. They consist of rules

and regulations, paperwork and guidance documents. Behind these lie norms and practices that are less explicit, but which govern such matters as how guidance is interpreted or how lenient to be in a given situation. Although they differ in detail, the basic outline tends to be the same across universities. Students who make claims of mitigating circumstances are asked to complete forms that describe their distress and outline its disabling qualities. They are asked to provide evidence, such as medical notes, scans or x-rays, to support their claim. The forms are then assessed by members of staff who can decide to grant an extension or not grant an extension according to rules and guidance, interpreted according to local norms and practices. These decisions are documented in standardised ways. There tends to be a higher-level committee that can adjudicate on difficult cases and deal with appeals. One anticipated outcome of these arrangements is that students experiencing health problems or life crises are protected from additional pressures. Grades do not suffer because of disability, and stressors are mitigated. Academics who might sometimes fail to pay attention to their students or who do not grasp quite what students are going through are corrected by procedure. But, the intended benefits are wider than that. The university culture is shifted, nudged towards structural compassion.

It sounds like a cause for celebration. However, the social science literature on bureaucracy suggests reason for caution. When bureaucratic instruments like mitigating circumstance arrangements are introduced to an organisation, they often replace less specified or formalised arrangements. By definition, these earlier arrangements are difficult to investigate. In some cases, it might be that no adjustment to deadlines were permitted. In other cases, adjustments may have been allowed, but the decision making might have been entirely up to the judgement of individual members of staff. Equally, the judgement of staff might have been structured or constrained in some way, for example, by placing limits onto the range of decisions that can be made, whilst leaving staff members with considerable latitude. In all cases, the introduction of bureaucratic instruments means the introduction of rules and procedure. Rather than relying on the judgement of staff members, rules and guidance, coupled with localised understandings of how to interpret those rules and guidance, shape what happens. Decision making becomes more standardised, based on explicit rationales that make the process transparent and, ideally, something close to predictable. If it seems impersonal, it is also unbiased. An organisation that is highly bureaucratised can be held accountable. If something goes wrong, there are (or should be) legible paper trails, documenting decisions made and the reasons for those decisions. Standardised paperwork makes it possible to systematically compare organisations. This can be helpful for quality monitoring, or if organisations are to be ranked or to compete.

There is a large (and critical) social science literature on bureaucracy. Bureaucratic instruments are widely seen as reductive, even, according to David Graeber, 'stupid' [14]. In many cases, bureaucratic instruments emerge as blunt objects, having a host of unforeseen effects whilst only minimally, if at all, engaging with the problems they were designed to address. They are often subverted or 'gamed' by those with the skills and knowhow. Michael Lipsky argued that these effects can sometimes be reduced by autonomy exercised by officials [15]. He described how 'street level bureaucrats' such as police or social workers have to follow the rules, but, in practice, exercise discretion about which rules to follow and how to interpret them. For Lipsky, simple rule following alone is insufficiently nimble, not flexible enough to adequately deal with complex human problems. But, the wriggle room provided by discretion compensates for this and enables improvisation and responsiveness, so that bureaucrats are able to adapt their work to meet the needs of individual cases. In practice, discretion also produces a kind of breathing space, allowing street-level bureaucrats to deal with an excessive burden of work, inadequate resources and limited time. For Lipsky, discretion has a negative side as well. It can create systemic biases, where officials consistently favour some populations (such as articulate and assertive middle-class people) and disadvantage others (such as minority or marginalised people). Henry Mintzberg and Alexandra McHugh coined the term 'adhocracy' to refer to organisations that give considerable autonomy to staff members to make the institution flexible and responsive [16].

An adhocracy might seem like the best of both worlds, accountable yet humane, guided by rules but not dominated by them.

But, it may be that producing rules is not the best way of achieving the desired effects. Erin Metz McDonnell suggests that what she calls 'pockets of bureaucratic effectiveness' arise from distinct subcultures that break off and form a separate identity, rather than from the imposition of centrally conceived rules [17]. Such an argument would count against the centralising logic of mitigating circumstance arrangements. And, by extension, we might worry that the institutionalisation of compassion is intrinsically problematic. For example, Lisa Stevenson suggests that care is necessarily spontaneous and open-ended, a human connection that involves seeing people in their uniqueness [18]. It cannot be specified by protocols. Rules and regulations might direct officials towards acting as if they care, but not actually make them care. So, structural care might even be an oxymoron, a kind of administrative pipe dream in which bureaucratic working appears to make the university compassionate in a rational and accountable way, whilst simultaneously eroding and ultimately erasing the meaningful human connections that constitute compassion.

We are thus presented with unsettling questions. Do mitigating circumstance arrangements help the right students in the right ways? How should we weigh the capacity of rules to overcome personal biases and make institutions accountable and comparable with the intuition that discretion is essential or that care is necessarily personal and spontaneous? We do not know if structural compassion is a coherent concept. What if bureaucratic instruments disrupt relationships; diminish opportunities for human connection; and create a more strategic, narrowly self-serving community?

An empirical investigation of mitigating circumstance arrangements is clearly demanded. There are, however, methodological problems to be negotiated. In official documentation, poor implementation can get buried, whilst apparent successes are foregrounded. For staff members, disclosing doubts or concerns can seem disloyal, unprofessional and even risky. Ethnographic methods provide an opportunity to hear how institutional processes are put into practice and to understand how staff members perceive these systems. Our study thus focuses on ethnographic interviews with trusted individuals with good knowledge of mitigating circumstance arrangements.

## 2. Materials and Methods

### 2.1. Design

This paper is based on ethnographic interviews with eleven individuals, conducted by NA. These were informal and unstructured, replicating a natural conversation as much as possible [19]. No recordings were made, but NA took contemporaneous notes. Where necessary, follow-up interviews were arranged to clarify details and to check shared understandings.

Both authors are active members of a university teaching community, with direct responsibility for academic mentoring and regular interactions with mitigating circumstance procedures. Mitigating circumstances are a hot topic on campus. Day-to-day life as an academic can mean participating in many conversations with university staff members about mitigating circumstance arrangements. So, even before the research began, we were familiar with multiple examples of mitigating circumstance appeals and benefitted from a broader sense of how university staff talk and feel about these issues. In particular, we were already aware of an important distinction between what staff members say in public and what they reveal in private. Our positionality is an instance of ethnographic immersion, an epistemic asset highly valued in anthropological research [20,21]. It gives our research methods a form of ecological validity [22]. We do not directly report on personal experience or prior conversations with colleagues and friends in this paper, but they helped us contextualise the interviews, provided an opportunity to test ideas and themes raised in interviews and guided our interpretation.

Recruitment was ad hoc and improvised, starting with existing academic contacts. NA employed a snowball technique to identify people with a deep knowledge of mitigating

circumstance arrangements. Some participants were known personally to NA; others were not. In some cases, NA followed up on particularly interesting or illuminating exchanges with colleagues regarding the nature of campus life in general or mitigating circumstance arrangements in particular. Our recruitment strategy has a number of methodological advantages. First, it enables a researcher to identify good participants: people who are interested in, and who have reflected on, the topics under discussion. Second, by building on existing relationships, it affords more open, trusting and exploratory dialogue. In Sarah Pink's terms, it enabled interviewer and interviewee to come together to 'create a shared place' [23]. This is particularly helpful when approaching a contentious and emotionally activating subject. Third, knowing something of the background of the research participant helped us situate the interview.

The aim of this paper is to investigate mitigating circumstance arrangements. This means the purpose of the interviews was not to understand staff experiences in and of themselves but to draw on staff observations and insights to better understand the arrangements. The participants are, thus, something like expert witnesses. They are not intended to be a representative sample and we do not measure the frequency of topics raised in the interviews. This research thus has a tight aim, samples from a specific and defined group and is motivated by established theory. We are not looking to make formal comparisons between groups or to conduct a cross case analysis. We believe our sample size is more than sufficient to give adequate information power [24]. The priority throughout is on data quality rather than quantity.

### 2.2. Participants

All names have been changed and efforts have been made to conceal identities. The eleven interviewees come from five universities, four in London and one in the south of England. All the universities included in the study are research universities with at least 10,000 students. One is focussed on the creative arts.

Andre is an administrator, who has worked in the HE sector for about fifteen years.

Andrew is an academic with five year's experience of teaching and research.

Elaine is a mental health professional who has worked for twenty years in various capacities for a university. She has also worked in the NHS.

Elizabeth is an academic with about five year's experience, whose role includes responsibilities related to mitigating circumstances.

Francis has recently retired from a university after a long university career.

Jillian is an early career academic whose role includes both welfare and research. She has been working in a university for about three years.

Paolo is an early career academic whose role combines teaching and admin. He completed his PhD a little over a year ago.

Phillipa is an academic with about fifteen years' experience in both teaching and research, ten of which were in her current university.

Triv is a mental health professional who has worked for a university for two years. She also practices privately and has worked in the NHS.

Raj works in university HR. His responsibilities include mitigating circumstances.

William is an academic with a career of over thirty years, across several universities.

### 2.3. Data Collection and Analysis

Several of these participants shared with NA particular mitigating circumstance cases. Most participants had reflected on the mitigating circumstance arrangements and the wider bureaucratic environment of the university, in many cases drawing on their own disciplinary expertise. A historian analysed the arrangements in historical terms, and a therapist focussed on how bureaucratic working impacts relationship dynamics. The interviews included both participant experiences of the arrangements and their reflections and theories about the arrangements.

This research was conducted according to the disciplinary norms of social anthropology [20]. In the ethnographic interviews, the aim was both to collect pre-formed opinions and ideas from participants and to engage in further exploration and reflection. The interviews were all reciprocal, improvised, creative and collaborative. In this sense, they might be seen as instances of coproduction. Because of this, we did not make a sharp distinction between data collection and the process of analysis. In our work, we directly drew on the way our participants analyse mitigating circumstances to produce our own analysis.

Subsequent analysis and authorship rests with the researchers [25]. The interviews were analysed according to anthropological methods [26]. Specifically, notes taken during and after the interviews were read and reread. Key themes were identified. Particular attention was paid to matters of tone, and sensory and embodied aspects of the interviews. Themes were identified through a process of reading and re-reading notes, reflecting on these and organising ideas emerging from reflection. Ethnographic working through this process is inherently comparative [27]. Thus, comparisons were made between interviews, between interviews and the researchers own experiences, and between interviews and published materials. Theory was occasionally used to illuminate or further develop the argument.

## 3. Results

All participants thought that the students were troubled. Francis, who had recently come to the end of a distinguished career remarked: 'You know, I wouldn't make it today, in academe, I couldn't manage the pressures, the stresses'. Participants described students as unhappy, lonely, anxious, unfulfilled, over-cautious and joyless. It could sound rather bleak, a story of hapless, even vulnerable students, lost in a toxic environment. However, doubts about how students describe their distress were also widely discussed. This emerged as a kind of paradox: students are not thriving, but they also describe their lives in ways that distort, or exaggerate, or, at least, muddy the waters. William spoke about how students cultivate a kind of vulnerability. He said: 'they have adapted to the world we made for them, for the extensions and mitigations. They are smart'. Others were more critical of the students themselves. Some thought that students had become fragile and overreported problems that might otherwise be brushed off, or lived with. Several participants told NA that learning to deal with stressors and crises is part of what people come to university to learn.

However, it was not clear how well suited mitigating circumstance arrangements are in addressing these problems. When they speculated on the origins of student distress, participants primarily discussed wider cultural features of the campus. They spoke about technology, particularly social media as well as financial pressures, precarity and uncertainty. Some academics thought that the student workload was more arduous than in the past, with more deadlines and assessed pieces of work. But, this perspective was not shared by all. Other participants told me that courses had become simpler, teaching styles became friendlier and assessments were more lenient, meaning less was expected of students now than in the recent past. Elizabeth noted that several of her students face multiple pressures, with caring responsibilities, for example, eating into study time. Raj described how the expansion of the university sector meant new demographic groups were now attending university. 'It's a social experiment' he said. Some referred to a decline in social skills, particularly after the COVID-19 lockdowns. Others spoke of social habits, tastes and strategies being part of the problem. For Andrew, for example, the way students approach sociality fails to generate healthy communities. He suggested that the problems originate outside the campus, in unfolding generational changes that are deskilling young people.

The use of the kind of medicalised or mental health language demanded by mitigating circumstance arrangements met a mixed response. The university administrators, Andre and Raj, found wellness language natural and unproblematic, simply a scientific way of representing distress that enabled evidence-based institutional responses. The mental health professionals, Elaine and Triv, were a little more circumspect. They both discussed

how 'having anxiety' could be problematic, as it blurred the line between mental disorders like bipolar and everyday distress. And, they both speculated that some students adopt mental health labels because they shield an individual from other descriptions that might be more wounding. As Triv put it: 'Who wouldn't want to be depressed rather than unpopular, or anxious rather than undesirable?' Some staff were sceptical about, even hostile towards, medicalised or psychologised language. 'Wellbeing' in particular was subject to some critique, even mockery. 'Wellbeing is unimportant and we are doing the kids a disservice by promoting it' said Jillian. There was wide discussion of what was felt to be a generational change in attitudes to labels. Most participants remarked that they were surprised by, sometimes baffled by, how comfortable students are with labels that they personally find problematic and disempowering.

Several participants referred back to their own student days, recalling fewer formal measures to deal with distress and disability. At least in their recollection, nobody saw the university in general as particularly interested in their welfare. There were few institutional provisions and little public messaging about wellbeing, flourishing or mental health. When they arrived at university, they had to learn how to deal with problems themselves. But, many participants remarked on how they had personally benefitted from relationships with particular staff members. William recalled a wonderfully helpful and supportive relationship with an inspiring tutor. Andrew remembered having dinners with a lecturer and his family when going through a dark spell, something that he said he would never do for his students today. Elaine described in detail how, early on in her career as a practitioner on campus, she had time to simply meet and chat with students. 'It would be unimaginable today' she said, 'to just be with students'.

None of the participants expressed confidence in the mitigating circumstance system. It was regarded as an open secret that there is a gap between ambition and reality, ideals and implementation, and policy and practice. It was an emotive topic that people wanted to talk about, even vent about, provided what they said could not be traced back to them. Philippa spoke with me at length about her concerns, meeting several times to try to make sense of what is going on. She is painfully aware of student distress but deeply frustrated at the institutional response. For the first five years of her career, she taught at a university where mental health was less prominent and where it was largely left to her discretion. She now teaches at a university where, after an institutional shake up, it is a campus-wide priority. Phillipa's current role includes a lot of responsibility for dealing with student mental health issues, especially extensions on deadlines. She said that the drive for institutionalised measures promoting mental health has come from student pressure and, in her words, 'from on high' because the university management feel they are not competing as successfully as they could with other universities and this was creating financial risks. Part of the problem, she explained, is that on paper, the system might look like it is working. Extensions are granted to people who need them. But, there is a deceptive side to this. For Philippa, the real work of the system is promotional. It is designed to secure better reports from the National Student Survey, so that the university attracts plenty of applicants.

Philippa sees the transition towards a greater emphasis on student mental health through bureaucratic instruments as a transition from personal discretion and judgement to the enactment of rules and regulations. She finds that enacting mitigating circumstance arrangements constrains her autonomy, leaves her with limited room for manoeuvre and inhibits her capacity to pay attention to individual students. Like other of my participants, Phillipa felt that her relationships with distressed students are adversely impacted by the mitigating circumstance arrangements. She described how she has to listen with half a mind to what the rules say and what decisions she is authorised to make. As a result, conversation becomes formalised and rather stilted, less spontaneous and less genuine than she would have preferred.

The mitigating circumstance arrangements, in Philippa's view, do not work. She thinks that many student problems are caused by financial pressures, family problems, and a combination of paid work and academic work being too much. In her experience,

this disproportionately affects minority students. But, the system does not reach them. She knows of particular students dealing with immense pressures at home, struggling with multiple jobs or dealing with major health problems who just do not engage with the system. They are too busy, or too depressed, or, somehow, find that the language around mental health just does not appeal. Instead, at least in Philippa's experience, the system has been co-opted by groups of vocal, confident, privileged students who know how to work the system. Phillipa described it as 'middle class female students' that 'hold us hostage'. They use it to secure more time for coursework but have only very normal levels of anxiety and depression. Administrators Raj and Andre both made similar points. They concurred that the system seems to be rewarding those who are good at negotiating bureaucracy, and those students are typically not the ones for whom the system is designed.

A common topic of conversation among participants was how mitigating circumstance arrangements medicalise and individualise problems. To illustrate this, one participant told me about Abeni, who comes from a wealthy Nigerian family. Abeni went to boarding school in England for all of secondary school. She is bright, cheery, energetic and thoughtful. But, when asked if she likes university, Abeni stated that she is unhappy and feels university is not for her. She has made several mitigating circumstance requests based on depression and ADHD, which she is able to evidence from GP e-consult forms. But, in person, she explains that she actually feels quite upbeat and enthusiastic when not in university classes. She has been offered CBT but does not seek further support for her mental health. Abeni never refers to race or ethnicity in connection to her wellbeing. But, the participant told me that Abeni is the fourth black student (out of six in her cohort) to state the same thing. The participant put it succinctly: 'For obvious reasons: CBT will not help to alleviate structural racism'.

Several participants discussed the emotional labour required of students when they pursue mitigating circumstance claims. As Elizabeth put it: 'when students are genuinely distressed, asking them to seek out and provide formal evidence is often time-consuming and a source of further distress'. She illustrated this by referring to a student of hers called Mikhail, who she sees as exceptionally gifted, one of the top in his cohort. Mikhail is Ukrainian, and he submitted a mitigating circumstance claim because of Russian rocket strikes near his family. He was unable to produce evidence to support the claim. Elizabeth believed him and would have been happy to approve the claim, but the rules prevented this. If he wanted to, Mikhail could have asked for the matter to be settled at a higher level, by a faculty committee. Instead, he chose to complete the assignment on time.

Elizabeth noted the institutional logic. She knows Mikhail, but the higher-level committee does not. They are in a less good position to make an informed decision about his case, but, in virtue of their seniority, are structurally better able to deal with responsibility. This, she suggested, revealed a tension in the system between caring for students and being institutionally accountable. The system, she thinks, prefers less well-informed decisions made by groups of more senior people over better-informed decisions made by less senior people because, ultimately, being accountable is more important than being compassionate. She added that she saw that the rules harmed her relationship with Mikhail. She regards Mikhail as trustworthy. But, she found herself being insincere with him, in that she felt she had to behave as if she supported the demands made by the mitigating circumstance rules.

Enacting the mitigating circumstance arrangement can lead to complex ethical challenges. Triv illustrated this with reference to a student called Kiko: 'Kiko has made multiple extension requests, and with each we get a better picture of her life'. She has a diagnosis of Emotionally Unstable Personality Disorder, self-harms, has a difficult family situation, and speaks regularly to the department's part-time in-house student counsellor. Kiko exemplifies two ethical challenges. First, she often evidences her mitigating circumstance claims using screenshots of personal or private communication, such as family texts. Triv commented that as evidence, these texts are much easier to provide than, for example, a medical certificate from her brother's hospitalisation, or evidence of the death of her best friend. But, Triv is uncomfortable. She does not know if these messages are shared with

the consent of the family members, and suspects they are not. A second ethical concern arises out of how information from the departmental counsellor becomes used in decisions regarding adjustments. According to the rules, this should not happen. But, informally, staff members talk. This breaches the confidentiality of the counselling sessions. As Triv put it: 'when she talks to her counsellor, she is also talking to us'. It also means that paperwork can be misleading because it does not contain all the evidence considered or the reasoning that led to a decision.

Several participants described how mitigating circumstance arrangements are simplistic and reductive. They fail to address the full complexity of student lives, yet because they seem to be the main concession that the university administration can grant, they gain a strange, disproportionate status. Jillian described them as being 'fetishized'. They seem to underscore the seriousness of assessment. Students appear constantly judged in all areas of their life. Paolo noted that as an undergraduate, just over a decade ago, he had been able to make social mistakes. Indiscretions at public events, for example, so much a part of what he saw as his student experience, went unnoticed, or if not, would soon be forgotten. No longer. Now, after social events, students compare photos and stories. Everything is recorded and collectively evaluated. The student body was framed as watchful, unforgiving and competitive. Triv spoke for many when she compared the light-hearted spontaneity of her student days with the deadweight of seriousness students are burdened with today. 'They are too scared to have fun' she said, 'everything matters so much'. For this reason, at times, it sounded as if they were almost part of the problem, not part of the solution.

Elizabeth gave an example of the limitations of mitigating circumstance arrangements by discussing at length her relationship with a student called Soren. She described Soren as intelligent and charming but also immature and disorganised. He took a year out for family and mental health problems during the COVID-19 pandemic, and his productivity seems continually to be marred by distress. Soren says he is not depressed. He apparently comes across as funny and cheerful in person and is enthusiastic and insightful in classes. After lengthy conversations, it seems that he finds university infantilising (especially many of the assessments). But, the problem seems existential. Soren says he does not have a goal or passion in life. His day-to-day existence lacks meaning. He says that he is tempted by religion as a way out of his sense of the pointlessness of life but knows nothing about it. Elizabeth saw his struggles as legitimate and his account as honest, but there is no option for 'existential crisis' in the list of approved reasons for extensions. As a result, she encouraged him to feign depression to secure mitigating circumstance approvals. This has led to him completing an e-consult form for his GP detailing depressive symptoms and then uploading a copy of that form when he applied for a mitigating circumstance extension.

Some participants warned me that negotiating the system has harmful effects on students. For example, I was told of a student called Khadija. It appears that Khadija is a very likable and able student but, over time, has become problematic for staff. She disappears for significant periods of time, during which she does not show up to lectures and does not reply to emails. Then, she books lots of meetings with tutors, without really having all that much to say in them. Khadija has acquired a range of diagnostic categories. This has, I was told, made Khadija's distress highly institutionally legible. It gets results. She is now in her third year. Over time, she has obtained extensions and a range of other accommodations. As a result, her life is now considerably less structured by deadlines than it would otherwise be. Khadija apparently complains of a lack of structure, a sense of being suspended in time and space. This might be a consequence of a mental disorder. But, it is also seems to be the effect of adjustments made in view of her disability. Through informal discussions, staff have found that she presents rather differently to different people. She appears at times to be less than fully candid. This was presented to me not meant as personal criticism, but as a form of adaptation. Khadija has become attuned to her institutional environment. However, my participant speculated about how helpful this will be in the longer run.

## 4. Discussion

The need for institutional care or institutional compassion is clear. All participants saw students as troubled. A single initiative like mitigating circumstances cannot, of course, be expected to be a panacea. But, there is a question about how well suited the arrangements are, relative to the problems they are intended to address. Is this the kind of institutional response that is needed? Indeed, as an instance of institutional vigilance, are the arrangements part of the ever-recording, unforgiving, bureaucratised world that seems to be associated with so much distress? Are these the sorts of arrangements that might promote structural compassion?

The literature on bureaucracy and accountability warns of inevitable and costly shortcomings. Bureaucratic working is simplistic, reducing complex goals to rudimentary procedures. Too crude for the complexity of human lives, accountable bureaucratic institutions are liable to be gamed by those with the skills and to have unforeseen and unwanted effects. All of these points were raised by the participants. Framing distress in medical terms is clearly reductive. Abeni appears to be suffering from structural racism, not depression. Soren, repackages a life crisis as mental disorder. Philippa, Raj and Andre all see gaming by middle class students redirecting the arrangements away from the people who need them and towards undeserving students who use them to boost their grades. The fact a more senior but less well-informed committee was empowered to decide Mikhail's case suggests that the primary goal of the arrangements is to take responsibility, not create compassionate relationships or caring decisions. Kiko's case demonstrates complex and unwanted ethical consequences.

Mitigating circumstance arrangements offer staff members limited scope for discretion. A request for an extension may be granted or not granted, certain kinds of reason are deemed justified; others are not. The whole process is monitored. It does not look like an adhocracy. However, in two cases, staff members told me they knowingly break the rules with the aim of helping a student. Kiko's departmental counsellor discloses confidential details to the member of staff responsible for mitigating circumstances. Soren's tutor appears to be colluding with him to misrepresent his distress. Philippa talked explicitly about her lack of discretion. She suggested that as well as constraining her decisions in unfortunate ways, it has a detrimental impact on relationships with students. This is because being aware of rules and regulations impairs attention and introduces an element of insincerity. In Berlant's terms discussed above, this sounds like the reverse of the 'more nuanced and capacious engagement'. In other words, this sounds like the opposite of structural compassion.

Participants appear to offer an ambivalent message on the appropriateness of mitigating circumstance measures. Some took the view that students face excessive academic pressures. We might reasonably think that mitigating circumstance arrangements have the capacity to reduce this. Others, however, thought that student work had become easier and that the teaching environment had become more supportive. When participants discussed the pressures on students, health and disability were mentioned but did not emerge as the most important factors. Having multiple jobs and living in an under connected, unforgiving and excessively socially competitive community seemed more significant. So, the fit between institutional measure and social problem seems partial. It might be argued that creating tiny breathing spaces for those suffering most from a toxic social environment is a way of facilitating structural dysfunction. For example, we saw that Abeni might be responding in a healthy way to structural racism, but this is relabelled in terms of individual pathology. In her case, the mitigating circumstance process appears to conceal structural problems, not to solve them. The reality of this problem is elaborated further by Nkasi Stoll, in her consideration of the impacts of structural racism in higher education on student mental health [27].

Philippa queried what the arrangements were really for, suggesting that they were an attempt to curry favour with students, so that improved National Student Satisfaction scores will lead to increased applications and so more revenue. Work by Sara Ahmed

on complaints procedures in universities offers support to Philippa's position. Ahmed describes how institutional arrangements can be cases of what she calls 'nonperformativity,' that is: 'speech acts that do not bring into effect what they name' [28]. Her point is that it is not just that the mechanisms fail or are ineffective. For Ahmed, these instruments can be an exercise in deception, a means of supressing complaints and maintaining the institutional status-quo whilst apparently doing the reverse. We do not see complete nonperformativity in our data. Mitigating circumstance applications are approved such that distressed people have their deadlines extended. But, Philippa perceives signs of the kind of smoke and mirrors that Ahmed writes about, in that she sees apparent institutional compassion as a form of marketing. Successful marketing might be rather different from institutional compassion. More insidiously, we might see mitigating circumstances as a way of medicalising students to avoid addressing failings in the institutional culture and environment. For some commentators, a tendency to focus on vulnerabilities in the individual at the expense of wider social and political factors is a characteristic feature of our times [29].

In the Introduction to this paper, we noted that Jonathan Haidt and Carol Gilligan see a dynamic relationship between compassionate communities and the individual members that make up that community [7,8]. They suggest that caring relationships help individuals become more empathic, for example, and that communities of empathic individuals promote collective flourishing. This suggests a different way of thinking about the campus and what a compassionate campus might entail. Individual students (and members of staff) are not just static entities that either have (or do not have) their needs and preferences met. Rather, they are changed by their social environment. Seen in this light, structural compassion might be a campus environment that impacts individuals in positive ways through the kinds of relationships that are created. However, when we look at the evidence provided by our participants, it appears that less favourable social environmental effects may also be in play. For example, several participants noted how students become strategic and perhaps a little disingenuous in how they frame and evidence their distress. The mitigating circumstance arrangements may be said to incentivise disability. To a casual observer, this can sound superficial, as if the distress is unchanged, just recorded in one way (as a mental disorder causing disability) rather than another (a student facing existential woes) for example. But, the social scientific literature suggests that these effects may run deeper, that individual and environment are more intimately entwined.

The philosopher Ian Hacking shows how medical descriptions of groups of people can, in his phrase, 'loop' back and change the people they describe, turning them into what he calls 'moving targets' [30,31]. In a sense, we are all moving targets, all shaped by the times we live in. But, Hacking is interested in how people may come to experience symptoms that are real, involuntary, measurable and treatable but, nonetheless, require certain social and cultural preconditions to occur. He suggests, for example, that multiple personality disorder cases tend to cluster around clinics that treat multiple personality because the treatment is something like a precondition of the disorder. People who experience themselves as multiple, as comprising of several discrete personalities that do not know each other, tend to do so under certain conditions, and, if Hacking is right, one of those conditions is mental healthcare directed at multiple personality.

If students are seen as moving targets, whose interior lives are shaped in part by institutional practice that loops back and changes them, we might want to think through what kinds of distress are being enabled by mitigating circumstance procedures and whether these effects are structurally compassionate. It may be that universities unknowingly propagate certain forms of distress through mitigating circumstance arrangements. The arrangements have multiple features that might be candidates for looping effects. These features include assumptions such as the following: if distress interferes with work, it is caused by a mental disorder; these mental disorders are likely to consist of experiences such as anxiety and depression (rather than, say, disenchantment or ennui); mental disorders are located in the individual in such a way that they may be apprehended by a medical

practitioner; and the best way a university might respond is by delaying deadlines. If looping effects are to be found on a campus, this might take the form of students finding themselves beset by troublesome emotions arising out of personal pathologies that fall under the expertise of medical professionals and which require a relaxation of deadlines. These might occur in lieu of existential crises, or stresses arising out of toxic social environments, both of which fall outside the expertise of medical professionals and neither of which are satisfactorily addressed by the relaxation of deadlines. In other words, if the arrangements were changed, distress might change too. We cannot explore the nature of looping effects on students here. Future research is needed, both to explore these issues empirically and to theorise their effects. But, here, we note the potential for looping effects arising out of mitigating circumstance arrangements that appear to contradict the goal of structural compassion.

Cheryl Mattingley and colleagues suggested a different way to theorise environmental causality that shows how it becomes entwined with an individual's own agency. They suggest that social situations can give rise to moral intuitions that might not be deterministic but can nonetheless be felt to be authoritative or binding such that, in conjunction with personal agency, they go on to shape life trajectory and self-formation [32]. They call this a 'moral engine'. This might be helpful in thinking about Khadija. The moral engine of university bureaucracy seems to encourage Khadija to dissemble. Her encounters with university bureaucracy may diminish her sense of the importance of telling a single, consistent self-narrative. Over time, perhaps, as she tells divergent, pragmatically motivated accounts of herself, in which expedience trumps candour, the capacity to do so will become eroded. Institutional accommodations successfully reduce Khadija's work pressures, but they might also have undermined her capacity to work to a deadline, even undermined her self-efficacy. In the longer run, short-term institutional fixes might be seen as morally highly questionable, impacting self-awareness, confidence and self-respect.

## 5. Conclusions

We started this paper by situating mitigating circumstance arrangements as 'structural care' or 'structural compassion', instances of the whole university approach. We argued that the arrangements might be assessed both in terms of their benefits to the students using the system, and in a wider way, as drivers of structural compassion, in which, piece by piece, the university is made into a healthier, more caring social world. The social science literature on bureaucracy gave us pause because it shows how inexact bureaucratic instruments are and how prone to misuse and unwanted consequences they can be. We also noted that the now burgeoning literature on care suggests that it might be difficult, perhaps even impossible, to bureaucratise, because of its immediate, human, spontaneous nature.

We then turned an ethnographic eye to eleven interviewees who discussed their doubts about the system. All expressed grave concerns about student mental health. The need for structural compassion was not in doubt. But, none were happy with the existing mitigating circumstance arrangements. The measures were presented as burdensome, imprecise and unduly medicalising, forcing students to relabel distress as mental disorder. In one case, the system appeared to conceal structural racism by presenting it as personal mental fragility in those suffering racism. In another, a staff member colluded with a student, telling him how to get medical evidence of a mental disorder that neither believed in. The measures as a whole emerged as vulnerable to co-option by assertive groups within the student population and failing to reach the students who really need them. Relationships between caring members of staff and students appear damaged, not enhanced, by the arrangements. Where conflicts arose between care and accountability, the system prioritises accountability. The intimate nature of the evidence presented led to various rather alarming ethical consequences. For some, the purpose of the arrangements is to create an impression of care, an instance of what Sara Ahmed calls 'nonperformativity'. The true aim, it was suggested, is to try to secure better student feedback and so increased applications.

Irrespective of the intentions of the designers of the mitigating circumstance arrangements, concerns were raised about the appropriateness of the measure and the effect they have on students. Indeed, at times, it was not completely clear whether the measures are part of the solution or part of the problem. Staff members saw the plight of students as arising out of broad social factors. Students were described as fearful and over-cautious, hampered in their pursuit of human connection by a sense of exposure to the judgemental eye of their peers. Post-COVID-19 student communities are thin and unsupportive, leaving students exposed and under-resourced. The mitigating circumstance arrangements ignore or, perhaps, erase these wider pressures, locating the problem in the person. Several participants contrasted the present university unfavourably with their own student days. Participants noted how in the past there seemed to be more time for staff and students to interact, leading to some rich, sustaining relationships. Today's academics find they do not have so much time, in part, ironically, because of the increased administrative load they carry and an increased sense of risk around students.

Encounters with bureaucratised care have the potential to be harmful to students. The mitigating circumstance arrangements appear to have the capacity to, in Ian Hacking's terms, 'loop' back and shape student distress. This means students are more likely to experience mental disorders and less likely to understand their problems in structural, cultural or social terms. Cheryl Mattingley's concept of 'moral engines' suggests that the arrangements might have harmful effects on students because they reward tactical but inauthentic disclosures. These issues require further investigation but clearly suggest capabilities and potentialities that are in tension with the broader goals of structural compassion.

Our evidence raises important questions about the capacity of institutions to enact ethical ideals through protocols, questions that are likely to become more important over the coming years. In 2022, the judge presiding over the legal case between Natasha Abrahart's family and the University of Bristol found that the university had discriminated against Natasha by marking her down in an oral presentation. The judge concluded that 'there was direct discrimination especially once the university knew or should have known that a mental health disability of some sort was preventing Natasha from performing'. Importantly, the judge took the view that the university *should* have known about student's mental health and have measures in place to respond appropriately. This ruling has real implications for how universities respond to student distress. In particular, the judge suggested that universities should address systemic problems by means of bureaucratic instruments, rather than relying on more ad hoc or improvised (but also more personal) processes. Our findings suggest reasons for caution in making such an argument. Bureaucratic instruments enable universities to document both that they care for students and that they respond appropriately to disability. However, we suggest that this documentation should not be taken at face value. This is because behind reassuring paperwork lie the multiple problems discussed above. It may be that pressures for more bureaucratic instruments, whether they arise out of student feedback; legal process; or, simply, good intentions, create a less compassionate campus, making it less likely that students thrive and, ultimately, adversely effecting student mental health.

**Author Contributions:** Conceptualization, N.A.; methodology, N.A. and N.C.B.; data collection, N.A.; formal analysis, N.A.; writing—original draft preparation, N.A.; writing—review and editing, N.A. and N.C.B.; funding acquisition, N.C.B. All authors have read and agreed to the published version of the manuscript.

**Funding:** This research was funded by Medical Research Council grant number MR/w002442/1.

**Institutional Review Board Statement:** Ethical clearance for this study was granted on 4 November 2022, reference number MRA-22/23-34521.

**Informed Consent Statement:** Informed consent was obtained from all subjects involved in the study.

**Data Availability Statement:** The data presented in this study are available on request from the corresponding author. The data are not publicly available due to ethical reasons.

**Conflicts of Interest:** The authors declare no conflict of interest.

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
