# Peer review of "The Impact of Mitigating Circumstances Procedures: Student Satisfaction, Wellbeing and Structural Compassion on the Campus"

_education, doi:10.3390/educsci13121230_

Round 1

Reviewer 1 Report

Comments and Suggestions for Authors

Pretty good article overall; methods section could be improved. There could be more information included regarding why the participants were selected (other than for convenience). Also, there could be more information included on the data analysis methods utilized for the study.

Author Response

We thank the reviewer. We have improved the methods section as suggested. Please note the text in yellow below. The new material discusses recruitment in more detail, including the rationale and advantages of the methods adopted. Additional material has been added outlining data analysis. The new text suggests that our analysis is fairly standard within anthropology. We provide six additional references to support this section. I hope the changes are satisfactory.

  1. Materials and Methods

This paper is based on ethnographic interviews with eleven individuals, conducted by NA. These were informal and unstructured, replicating a natural conversation as much as possible [19] No recordings were made, but NA took contemporaneous notes. Where necessary, follow up interviews were arranged to clarify details and check shared understandings.

Both authors are active members of university teaching community, with direct responsibility for academic mentoring and regular interactions with mitigating circumstances procedures. Mitigating circumstances are a hot topic on campus. Day-to-day life as an academic can mean participating in many conversations with university staff members about mitigating circumstances arrangements. So even before the research began, we were familiar with multiple examples of mitigating circumstances appeals, and benefitted from a broader sense of how university staff talk and feel about these issues. In particular, we were already aware of an important distinction between what staff members say in public and what they reveal in private. Our positionality is an instance of ethnographic immersion, an epistemic asset highly valued in anthropological research [A,B]. It gives our research methods a form of ecological validity [C]. We do not directly report on personal experience or prior conversations with colleagues and friends in this paper, but they helped us contextualise the interviews, provided an opportunity to test ideas and themes raised in interviews, and guided our interpretation.

Recruitment was ad-hoc and improvised, starting with existing academic contacts. NA employed a snowball technique to identify people with a deep knowledge of the mitigating circumstance arrangements. Some participants are known personally to NA, others are not. In some cases NA followed up on particularly interesting or illuminating exchanges with colleagues regarding the nature of campus life in general, or mitigating circumstances arrangements in particular. Our recruitment strategy has a number of methodological advantages. First, it enables a researcher to identify good participants: people who are interested in, and who have reflected on, the topics under discussion. Second, by building on existing relationships, it affords more open, trusting and exploratory dialogue. In Sarah Pink’s terms, it enabled interviewer and interviewee to come together to ‘create a shared place.’ (D] This is particularly helpful when approaching a contentious and emotionally activating subject. Third, knowing something of the background of the research participant helped us situate the interview.

The aim of this paper is to investigate mitigating circumstances arrangements. This means the purpose of the interviews was not to understand staff experiences in themselves, but to draw on staff observations and insights to better understand the arrangements. The participants are, thus, something like expert witnesses. They are not intended to be a representative sample and we do not measure the frequency of topics raised in interviews. This research thus has a tight aim, samples from a specific and defined group and is motivated by established theory. We are not looking to make formal comparisons between groups or to conduct a cross case analysis. We believe our sample size is more than sufficient to give adequate information power [E]. The priority throughout is on data quality rather than quantity.

All names have been changed and efforts have been made to conceal identities. The eleven interviewees come from five universities, four in London and one in the south of England. All the universities included in the study are research universities with at least 10,000 students. One is focussed on the creative arts.

xxx

Several of these participants shared with NA particular mitigating circumstances cases. Most participants had reflected on the mitigating circumstances arrangements and the wider bureaucratic environment of the university, in many cases drawing on their own disciplinary expertise. A historian analysed the arrangements in historical terms, a therapist focussed on how bureaucratic working impacts relationship dynamics. Interviews included both participant experiences of the arrangements and their reflections and theories about the arrangements.

This research was conducted according to the disciplinary norms of social anthropology [20]. In the ethnographic interviews, the aim was to both collect pre-formed opinions and ideas from participants, and to engage further to explore and reflect. The interviews were all reciprocal, improvised, creative and collaborative. In this sense, they might be seen as instances of coproduction, although subsequent analysis and authorship rests with the researchers [F]. The interviews were analysed according to anthropological methods [19]. Notes taken during and after interviews were read and reread. Key themes were identified. Particular attention was paid to matters of tone, and sensory and embodied aspects of the interviews. For many participants, the issue of mitigating circumstances is highly emotive, and we wanted to find ways to represent and theorise  this affective hinterland as well as particular instances of appeals. Ethnographic working is inherently comparative [20]. Comparisons were made between interviews, between interviews and the researchers own experiences, and between interviews and published materials. Theory is occasionally used to illuminate or further develop the argument.

Like any piece of research, our evidence is not independent of our method. A different researcher asking different questions would have written a different paper. But we suggest that our method is suited to the questions we ask, and that our findings are likely to be replicated by similar pieces of work.

[A] Davies, J Disorientation, Dissonance, and Altered Perception in the Field
Emotions in the Field : The Psychology and Anthropology of Fieldwork Experience, edited by James Davies, and Dimitrina Spencer, Stanford University Press, 2010. p79

[B] Sherman Heyl, B. (2001) ‘Ethnographic interviewing’, in P.Atkinson, A.Coffey, S.Delamont, J.Lofland and L.Lofland (eds) Handbook of Ethnography. London: Sage.p367

[C] O’Reilly, K. (2005) Ethnographic Methods. London: Routledge p115

[D] Pink, S. (2015). The sensoriality of the interview : rethinking personal encounters through the senses. In Doing Sensory Ethnography ( Second ed., pp. 73-93). SAGE Publications Ltd, https://doi.org/10.4135/9781473917057

[E] Malterud K, Siersma VD, Guassora AD. Sample Size in Qualitative Interview Studies: Guided by Information Power. Qual Health Res. 2016 Nov;26(13):1753-1760. doi: 10.1177/1049732315617444. Epub 2016 Jul 10. PMID: 26613970.

[F] Boyer, D and G E Marcus Collaborative Anthropology Today: A Collection of Exceptions. Ithaca a

Reviewer 2 Report

Comments and Suggestions for Authors

It has been a pleasure to review this paper, which illustrates a qualitative research very well done. I highlight especially the method, the defense of the authors, the quality of the interviews and interpretation of the results. 

Congratulations for the paper

Author Response

Thank you for a wonderfully supportive and encouraging review.

Reviewer 3 Report

Comments and Suggestions for Authors

Dear author and publisher. Studies from anthropology are fundamental for education, but I find very serious and severe flaws for which I recommend the rejection of this study. First of all, the sample is too small, and does not allow generalization, at least 30 people are needed, and being a generic field, since it is about European university students, the sample should be representative. It is not a minority group. Moreover, the time sequence is not indicated; it expects a longitudinal study of these students throughout their university studies. The way to proceed with the data is not explained either. 

Author Response

We thank the reviewer. We have improved the methods section to try to respond to their concerns. We have provided more detail on recruitment, data analysis and how this is situated within anthropology. Six new references connect our methods to the methods of other anthropologists. We have been more explicit than we were before that the sample is not intended to representitive, no group comparison is undertaken, and there is no longitudinal aspect to the research.  Please see the text in yellow below. We respectfully defend our methods as being a fairly typical example of anthropological research and note that another reviewer specifically praised our methods.

  1. Materials and Methods

This paper is based on ethnographic interviews with eleven individuals, conducted by NA. These were informal and unstructured, replicating a natural conversation as much as possible [19] No recordings were made, but NA took contemporaneous notes. Where necessary, follow up interviews were arranged to clarify details and check shared understandings.

Both authors are active members of university teaching community, with direct responsibility for academic mentoring and regular interactions with mitigating circumstances procedures. Mitigating circumstances are a hot topic on campus. Day-to-day life as an academic can mean participating in many conversations with university staff members about mitigating circumstances arrangements. So even before the research began, we were familiar with multiple examples of mitigating circumstances appeals, and benefitted from a broader sense of how university staff talk and feel about these issues. In particular, we were already aware of an important distinction between what staff members say in public and what they reveal in private. Our positionality is an instance of ethnographic immersion, an epistemic asset highly valued in anthropological research [A,B]. It gives our research methods a form of ecological validity [C]. We do not directly report on personal experience or prior conversations with colleagues and friends in this paper, but they helped us contextualise the interviews, provided an opportunity to test ideas and themes raised in interviews, and guided our interpretation.

Recruitment was ad-hoc and improvised, starting with existing academic contacts. NA employed a snowball technique to identify people with a deep knowledge of the mitigating circumstance arrangements. Some participants are known personally to NA, others are not. In some cases NA followed up on particularly interesting or illuminating exchanges with colleagues regarding the nature of campus life in general, or mitigating circumstances arrangements in particular. Our recruitment strategy has a number of methodological advantages. First, it enables a researcher to identify good participants: people who are interested in, and who have reflected on, the topics under discussion. Second, by building on existing relationships, it affords more open, trusting and exploratory dialogue. In Sarah Pink’s terms, it enabled interviewer and interviewee to come together to ‘create a shared place.’ (D] This is particularly helpful when approaching a contentious and emotionally activating subject. Third, knowing something of the background of the research participant helped us situate the interview.

The aim of this paper is to investigate mitigating circumstances arrangements. This means the purpose of the interviews was not to understand staff experiences in themselves, but to draw on staff observations and insights to better understand the arrangements. The participants are, thus, something like expert witnesses. They are not intended to be a representative sample and we do not measure the frequency of topics raised in interviews. This research thus has a tight aim, samples from a specific and defined group and is motivated by established theory. We are not looking to make formal comparisons between groups or to conduct a cross case analysis. We believe our sample size is more than sufficient to give adequate information power [E]. The priority throughout is on data quality rather than quantity.

All names have been changed and efforts have been made to conceal identities. The eleven interviewees come from five universities, four in London and one in the south of England. All the universities included in the study are research universities with at least 10,000 students. One is focussed on the creative arts.

xxx

Several of these participants shared with NA particular mitigating circumstances cases. Most participants had reflected on the mitigating circumstances arrangements and the wider bureaucratic environment of the university, in many cases drawing on their own disciplinary expertise. A historian analysed the arrangements in historical terms, a therapist focussed on how bureaucratic working impacts relationship dynamics. Interviews included both participant experiences of the arrangements and their reflections and theories about the arrangements.

This research was conducted according to the disciplinary norms of social anthropology [20]. In the ethnographic interviews, the aim was to both collect pre-formed opinions and ideas from participants, and to engage further to explore and reflect. The interviews were all reciprocal, improvised, creative and collaborative. In this sense, they might be seen as instances of coproduction, although subsequent analysis and authorship rests with the researchers [F]. The interviews were analysed according to anthropological methods [19]. Notes taken during and after interviews were read and reread. Key themes were identified. Particular attention was paid to matters of tone, and sensory and embodied aspects of the interviews. For many participants, the issue of mitigating circumstances is highly emotive, and we wanted to find ways to represent and theorise  this affective hinterland as well as particular instances of appeals. Ethnographic working is inherently comparative [20]. Comparisons were made between interviews, between interviews and the researchers own experiences, and between interviews and published materials. Theory is occasionally used to illuminate or further develop the argument.

Like any piece of research, our evidence is not independent of our method. A different researcher asking different questions would have written a different paper. But we suggest that our method is suited to the questions we ask, and that our findings are likely to be replicated by similar pieces of work.

[A] Davies, J Disorientation, Dissonance, and Altered Perception in the Field
Emotions in the Field : The Psychology and Anthropology of Fieldwork Experience, edited by James Davies, and Dimitrina Spencer, Stanford University Press, 2010. p79

[B] Sherman Heyl, B. (2001) ‘Ethnographic interviewing’, in P.Atkinson, A.Coffey, S.Delamont, J.Lofland and L.Lofland (eds) Handbook of Ethnography. London: Sage.p367

[C] O’Reilly, K. (2005) Ethnographic Methods. London: Routledge p115

[D] Pink, S. (2015). The sensoriality of the interview : rethinking personal encounters through the senses. In Doing Sensory Ethnography ( Second ed., pp. 73-93). SAGE Publications Ltd, https://doi.org/10.4135/9781473917057

[E] Malterud K, Siersma VD, Guassora AD. Sample Size in Qualitative Interview Studies: Guided by Information Power. Qual Health Res. 2016 Nov;26(13):1753-1760. doi: 10.1177/1049732315617444. Epub 2016 Jul 10. PMID: 26613970.

[F] Boyer, D and G E Marcus Collaborative Anthropology Today: A Collection of Exceptions. Ithaca and London: Cornell 2020

Round 2

Reviewer 3 Report

Comments and Suggestions for Authors

Dear Authors and Editor. You have made the changes in methodological issues but the data processing continues to be poor and has not been improved.

Author Response

We have added text in yellow to try to make the method of data analysis more explicit. We hope the reviewer finds this satisfactory.

This research was conducted according to the disciplinary norms of social anthropology [20]. In the ethnographic interviews, the aim was to both collect pre-formed opinions and ideas from participants, and to engage further to explore and reflect. The interviews were all reciprocal, improvised, creative and collaborative. In this sense, they might be seen as instances of coproduction. Because of this, we don’t make a sharp distinction between data collection and the process of analysis. In our work, we directly draw on the way our participants analyse mitigating circumstances to produce our own analysis.

Subsequent analysis and authorship rests with the researchers [F]. The interviews were analysed according to anthropological methods [19]. Specifically, notes taken during and after interviews were read and reread. Key themes were identified, with. Particular attention was paid to matters of tone, and sensory and embodied aspects of the interviews. Themes were identified through a process of reading and re-reading notes, reflecting on these and organising ideas emerging from reflection. Ethnographic working through this process is inherently comparative [20]. Comparisons were made between interviews, between interviews and the researchers own experiences, and between interviews and published materials. Theory is occasionally used to illuminate or further develop the argument.
